# High-Precision 3D Reconstruction Study with Emphasis on Refractive Calibration of GelStereo-Type Sensors

**DOI:** 10.3390/s23052675

**Published:** 2023-02-28

**Authors:** Chaofan Zhang, Shaowei Cui, Shuo Wang, Jingyi Hu, Yipeng Huangfu, Boyue Zhang

**Affiliations:** 1State Key Laboratory of Multimodal Artificial Intelligence Systems, Institute of Automation, Chinese Academy of Sciences, Beijing 100190, China; 2School of Artificial Intelligence, University of Chinese Academy of Sciences, Beijing 100049, China; 3Center for Excellence in Brain Science and Intelligence Technology, Chinese Academy of Sciences, Shanghai 200031, China; 4School of Future Technology, University of Chinese Academy of Sciences, Beijing 100049, China; 5Faculty of Information Technology, Beijing University of Technology, Beijing 100124, China

**Keywords:** visuotactile sensors, GelStereo-type sensors, refractive calibration, tactile 3D reconstruction, ray tracing

## Abstract

GelStereo sensing technology is capable of performing three-dimensional (3D) contact shape measurement under various contact structures such as bionic curved surfaces, which has promising advantages in the field of visuotactile sensing. However, due to multi-medium ray refraction in the imaging system, robust and high-precision tactile 3D reconstruction remains a challenging problem for GelStereo-type sensors with different structures. In this paper, we first propose a universal Refractive Stereo Ray Tracing (RSRT) model for GelStereo-type sensing systems to realize 3D reconstruction of the contact surface. Moreover, a relative geometry-based optimization method is presented to calibrate multiple parameters of the proposed RSRT model, such as the refractive indices and structural dimensions. Furthermore, extensive quantitative calibration experiments are performed on four different GelStereo sensing platforms; the experimental results show that the proposed calibration pipeline can achieve less than 0.35 mm in Euclidean distance error, based on which we believe that the proposed refractive calibration method can be further applied in more complex GelStereo-type and other similar visuotactile sensing systems. Such high-precision visuotactile sensors can facilitate the study of robotic dexterous manipulation.

## 1. Introduction

Tactile perception is one of the main ways for humans to interact with real-world environments [1,2], and has a natural appeal when visual impairment occurs. Naturally, endowing robots with human-like tactile perception capabilities has become an important part of robots moving towards the open reality [3,4,5]. Recently, visuotactile sensing technology has received increasing attention in the robotic tactile sensing community [6]. Compared to traditional tactile arrays [7] using sensing principles such as capacitive [8], resistive [9], and piezoelectric [10], visuotactile sensing technology has significant advantages in spatial resolution, cost, and stability, especially the ability to directly obtain high-precision tactile deformation [11]. To date, visuotactile sensors have realized the measurement of various tactile patterns such as sliding contact [12,13], six-axis force/torque [14], distributed force field [15], 3D contact deformation [16,17], etc. Furthermore, the acquired rich tactile sensing information significantly improves the robot’s dexterous manipulation ability, and enables the robot to successfully complete many challenging missions, such as USB interface assembly [18], cable manipulation [19], swing-up manipulation [20], in-hand manipulation [21], cloth manipulation [22], and more. In summary, the high-precision measurement of contact deformation by visuotactile sensors makes these complicated robotic dexterous manipulation tasks possible to a certain extent. In this sense, the visuotactile sensors can be called deformation sensors. However, it is a very challenging problem to realize high-precision 3D deformation measurement on arbitrary contact structures [23,24,25].

Currently, various types of tactile sensors have been developed, with different structures used to adapt to different sensing requirements and integration scenarios. A detailed survey of visuotactile sensing technologies can be found in [6]. Nevertheless, certain sensors cannot achieve high-precision 3D contact deformation measurement, such as FingerVision [26,27], TacTip [28,29], etc. To the best of our knowledge, there are four types of technical routes for visuotactile sensing technology that can achieve high-precision 3D contact deformation sensing. First, the GelSight-type sensors proposed by Yuan et al. [11,30] realize 3D reconstruction of the contact surface using photometric stereo algorithm. At present, the GelSight-type sensors are mostly integrated with the fingertip of parallel grippers [31]. Romero et al. extended the GelSight pipeline to a curvature contact structure, although the accuracy of 3D reconstruction is reduced by fitting the surface with a plane [32]. Second, Do et al. proposed the learning-based 3D reconstruction pipeline using monocular depth estimation, which is adopted by the DenseTact sensors [24,33]. However, the generalization ability of this learning-based route to various contact scenarios in practical real-world situations requires further consideration. Third, Soft-bubble sensors [34,35] directly employ the tailored depth sensors to measure 3D contact deformation, which is difficult to integrate in a setting as small as a bionic fingertip.

Recently, our previous studies proposed the GelStereo route based on a binocular stereo vision system [14,17,21]. The biggest advantage of this method is that there are no requirements for the contact surface structure, and it can achieve high-precision 3D contact deformation measurement on multi-curvature bionic contact surfaces. Due to the multi-medium refraction of light in GelStereo-type sensors, binocular stereo 3D reconstruction methods need to be remodeled; however, few studies have explored this issue. In [21], we used a regression method to directly realize high-resolution 2D-to-3D mapping (3D reconstruction), which requires a professional calibration platform to obtain enough ground-truth 2D-to-3D samples. Ma et al. presented a binocular visuotactile sensor with ray tracing modelling [36]. However, both the contact and refracting surfaces in their scene were flat, and the absolute 3D reconstruction errors were not fully evaluated.

In this paper, we carry out an in-depth study of the universal 3D reconstruction pipeline of GelStereo-type sensors, with an emphasis on refractive calibration. To begin with, a Universal Refractive Stereo Ray Tracing model, which we call GU-RSRT, is presented for GelStereo-type sensors with various structures. The GU-RSRT performs ray tracing modeling on the GelStereo imaging systems, in which the parameters include intrinsic and extrinsic parameters of the binocular camera, refractive indices, and structural geometry. To obtain these parameters in GU-RTST model, we propose a Universal Multi-Medium Refractive (UMMR) calibration method using the embedded relative geometric features of checkerboards. Furthermore, a Marker-Based Self-Calibration (MBSC) method is proposed for specific GelStereo-type sensors with known structured markers embedded on the sensor surface, which omits the step of checkerboard calibration and updates the sensor parameters during daily use, significantly improving the service life of the sensor. Extensive calibration and evaluation experiments are performed on four GelStereo-type sensors with different refracting and contact surfaces. The experimental results show that the proposed refractive calibration method can obtain reasonable parameters for the GU-RSRT model, and the constructed 3D reconstruction system achieves 3D points measurement with less than 0.35 mm Euclidean distance error on different sensor platforms. Furthermore, the proposed 3D reconstruction pipeline with refractive calibration can be practically applied to high-precision 3D deformation measurement in various GelStereo-type sensors and similar visuotactile sensors, including those with binocular cameras and undergoing multi-medium light refraction, as shown in Figure 1.

In summary, the contributions of this paper can be summarized as follows:A universal refractive stereo ray tracing model that can handle sensors with arbitrary refracting and contact surfaces is presented for GelStereo-type sensors.A universal multi-medium refractive calibration method using the embedded geometric features of checkerboards is proposed to obtain refractive parameters in GelStereo imaging systems. The results show that the proposed calibration method can realize high precision (less than 0.35 mm Euclidean distance error) in 3D contact geometry measurements on different sensor platforms.A marker-based self-calibration method that can automatically perform refractive calibration every time the sensor starts up is proposed for specific GelStereo-type sensors with known structured markers embedded on the sensor surface, allowing for prolonged sensor life.

The rest of this paper is organized as follows. We first provide a tactile 3D reconstruction pipeline for GelStereo-type sensors using a GU-RSRT model (Section 2). The refractive calibration methods are proposed to obtain the parameters of GU-RSRT model in Section 3. Then, the experimental design and results are presented in Section 4 and Section 5. Finally, we discuss the limitations and future works in Section 6, and conclude the paper in Section 7.

## 2. 3D Reconstruction of GelStereo-Type Sensors

GelStereo-type sensors utilize a binocular vision system for 3D geometry sensing. With sparse or dense stereo matching points pairs on left and right tactile images [17,21], a 2D-to-3D model is needed to reconstruct the 3D tactile points with high precision. The 2D-to-3D modeling using ray tracing method has shown good performance in previous work [37]. In this paper, we present a Universal Refractive Stereo Ray Tracing model that can be instantiated to any GelStereo-type sensors. The GU-RSRT geometrically models the propagation paths of light rays through multi-medium refraction in GelStereo-type sensor imaging systems, assuming that each medium is homogeneous (i.e., the rays propagate in a straight line in each medium). Figure 2 diagrams the GU-RSRT model in detail; key symbols are listed in Table 1. Without loss of generality, we introduce the GU-RSRT model with *m* times refraction and use the general equation $f_{*}\left(x,y,z\right)=0$ to describe the refracting surfaces.

### 2.1. Light Ray Path

Taking the left ray as an example, we backpropagate the rays from the camera optical center to the 3D points on the sensor surface in the left camera coordinate system.

Given a point on the left tactile image (indicated by $\left(u_{l},v_{l}\right)$), the direction of the ray from the left camera optical center is computed according to the pin-hole camera model:(1)$$\begin{array}{c} {\vec{r}}_{0}^{l}=\frac{K^{-1}{\left[u_{l},v_{l},1\right]}^{T}}{\left|\right|K^{-1}{\left[u_{l},v_{l},1\right]}^{T}{\left|\right|}_{2}}=\left(r_{0x}^{l},r_{0y}^{l},r_{0z}^{l}\right) \end{array}$$
where $K\in R^{3\times 3}$ indicates the intrinsic parameter of the camera and ${\vec{r}}_{0}^{l}$ denotes the direction vector of left ray in medium 0 (i.e., the air). The starting point of the ray is the camera optical center, indicated by $O^{l}\left(x_{0}^{l},y_{0}^{l},z_{0}^{l}\right)$. The equation of the left ray in medium 0 (short for left ray 0) is expressed as follows:(2)$$\begin{array}{c} \frac{x-x_{0}^{l}}{r_{0x}^{l}}=\frac{y-y_{0}^{l}}{r_{0y}^{l}}=\frac{z-z_{0}^{l}}{r_{0z}^{l}}. \end{array}$$

For rays in each medium *i*, we first solve the starting point ($P_{i}^{l}$), then compute the normal vector (${\vec{n}}_{i}^{l}$) of the refracting surface at this point, and finally infer the direction vector (${\vec{r}}_{i}^{l}$) of the ray according to Snell’s law:

The equation of the refracting surface between medium i−1 and *i* can be expressed as:(3)$$\begin{array}{c} f_{i}\left(x,y,z\right)=0. \end{array}$$

The solution to the simultaneous equations, including refracting surface equation $f_{i}$ (Equation (Equation 3)) and left ray i−1 equation, is the starting point of left ray *i*, indicated by $P_{i}^{l}\left(x_{i}^{l},y_{i}^{l},z_{i}^{l}\right)$. For example, $P_{1}^{l}\left(x_{1}^{l},y_{1}^{l},z_{1}^{l}\right)$ is solved by Equations (Equation 2) and (Equation 3) with i=1. Next, the normal vector of the refracting surface at point $P_{i}^{l}$ is computed by
(4)$$\begin{array}{c} {\vec{n}}_{i}^{l}=\frac{▽f_{i}{\left(x,y,z\right)|}_{P_{i}^{l}}}{\left|\right|▽f_{i}{\left(x,y,z\right)|}_{P_{i}^{l}}{\left|\right|}_{2}}. \end{array}$$

From [38], the direction vector of the left ray in medium *i* can be formed as follows:(5)$$\begin{array}{c} {\vec{r}}_{i}^{l}=\alpha {\vec{r}}_{i-1}^{l}+\beta {\vec{n}}_{i}^{l}=\left(r_{ix}^{l},r_{iy}^{l},r_{iz}^{l}\right) \end{array}$$
where $\alpha =\mu_{i-1}/\mu_{i}$ and
(6)$$\begin{array}{c} \beta =\sqrt{1-{\left(\frac{\mu_{i-1}}{\mu_{i}}\right)}^{2}\left[1-{\left({\vec{r}}_{i-1}^{l}\cdot {\vec{n}}_{i}^{l}\right)}^{2}\right]}-\frac{\mu_{i-1}}{\mu_{i}}{\vec{r}}_{i-1}^{l}\cdot {\vec{n}}_{i}^{l}. \end{array}$$

Then, the equation of the left ray in medium *i* can be expressed as
(7)$$\begin{array}{c} \frac{x-x_{i}^{l}}{r_{ix}^{l}}=\frac{y-y_{i}^{l}}{r_{iy}^{l}}=\frac{z-z_{i}^{l}}{r_{iz}^{l}}. \end{array}$$

Using Equations (Equation 1)–(Equation 7), the paths of rays from the left camera’s optical center to medium *m* are computed in the left camera coordinate system. In the same way, we obtain the trajectories of rays passing through the right camera optical center in the right camera coordinate system.

### 2.2. Intersection Points

The left and right rays intersect in medium *m* on the sensor surface. This intersection point is considered as the reconstructed 3D point. In order to compute this point, we transform the right ray *m* from the right to the left camera coordinate system:(8)$$\begin{array}{cc} {}^{L}P_{m}^{r} & ={{}^{L}R}_{R}{}^{R}P_{m}^{r}+{{}^{L}T}_{R}=\left(x_{m}^{r},y_{m}^{r},z_{m}^{r}\right) \end{array}$$(9)$$\begin{array}{cc} {}^{L}{\vec{r}}_{m}^{r} & ={{}^{L}R}_{R}{}^{R}{\vec{r}}_{m}^{r}=\left(r_{mx}^{r},r_{my}^{r},r_{mz}^{r}\right) \end{array}$$
where ${{}^{L}R}_{R}\in R^{3\times 3}$ and ${{}^{L}T}_{R}\in R^{3\times 1}$ denote the rotation matrix and translation vector between the right and the left camera coordinate systems, respectively. Then, an equation set to solve the intersection point is built as follows:(10)$$\begin{array}{c} {}^{L}P_{m}^{l}+t^{l}{}^{L}{\vec{r}}_{m}^{l}={}^{L}P_{m}^{r}+t^{r}{}^{L}{\vec{r}}_{m}^{r}\Rightarrow \left\{\begin{array}{c} x_{m}^{l}+t^{l}r_{mx}^{l}=x_{m}^{r}+t^{r}r_{mx}^{r}\\ y_{m}^{l}+t^{l}r_{my}^{l}=y_{m}^{r}+t^{r}r_{my}^{r}\\ z_{m}^{l}+t^{l}r_{mz}^{l}=z_{m}^{r}+t^{r}r_{mz}^{r} \end{array}\right.. \end{array}$$

This equation set is overdetermined; thus, that the least square method is employed. The reconstructed 3D point *P* can be computed by
(11)$$\begin{array}{c} P=\frac{{}^{L}P_{m}^{l}+{\hat{t}}^{l}{}^{L}{\vec{r}}_{m}^{l}+{}^{L}P_{m}^{r}+{\hat{t}}^{r}{}^{L}{\vec{r}}_{m}^{r}}{2} \end{array}$$
where (${\hat{t}}^{l},{\hat{t}}^{r}$) is the least square solution of Equation (Equation 10).

## 3. Refractive Calibration

### 3.1. Problem Formulation

To achieve high-precision 3D geometry sensing, a refractive calibration method is desirable in order to obtain a fine set of parameters for GU-RSRT model. The model parameters are divided into three parts, including camera parameters, refractive indices, and structural parameters. Zhang’s method [39] is employed to obtain the intrinsic and extrinsic binocular camera parameters using checkerboard images taken in the air. How to calibrate the parameters of refractive indices and structure is the primary task of this section.

In GelStereo-type sensor imaging systems, the shapes of refracting surfaces are already known, as they are determined by the sensors’ structure design. However, the pose of the refracting surface in the camera coordinate system is uncertain due to deviations during sensor assembly. The pose is determined by several translation and orientation parameters (indicated by $\phi_{s}=\left\{\cdots \right\}$), which vary on various sensor platforms, mainly related to the characteristics of the refracting surface. In addition, the refractive indices (indicated by $\phi_{r}=\left\{\mu_{0},\mu_{1},\cdots ,\mu_{m}\right\}$) are determined by material properties, which are affected by the components and molding environment. In summary, the parameter sets for the structure $\phi_{s}$ and refractive indices $\phi_{r}$ are unknown and require calibration.

Considering the reconstructed 3D points *P* using the GU-RSRT model, the refractive calibration is defined as an optimization problem:(12)$$\begin{array}{c} \phi_{s},\phi_{r}=\arg\min_{\phi_{s},\phi_{r}}\;F\left(P\right) \end{array}$$
where *F* is the objective function. Then, how to build this objective function becomes the main problem in calibration.

### 3.2. Universal Multi-Medium Refractive Calibration Method

In GelStereo-type sensor imaging systems, we observe that 2D-to-3D models that do not take refraction into consideration or have poor refraction parameters could lead to distortion of the reconstructed 3D point clouds. Our main idea for calibration is to optimize the parameters of the refraction system by reconstructing undistorted 3D point clouds. Objects or patterns with known structures can provide the ground truth of relative geometric features. Among these, checkerboards with known corner point structures are easy to make and detect. Here, we propose a Universal Multi-Medium Refractive (UMMR) calibration method for GelStereo-type sensors. In practice, we fully pressed checkerboards onto the surface of the sensor’s transparent elastomer. The 3D point of each corner was then reconstructed by the GU-RSRT model. The objective functions were designed to ensure the spatial invariance of the checkerboard corners.

As shown in Figure 3, the Euclidean distance and perpendicularity can be used to constrain the geometric relationship of the checkerboard corners. The number of checkerboard corners is M×N, the length of each cell is denoted by *l*, and $C_{i,j}$ indicates the corner point at the i-th row and the j-th column. The ground truth Euclidean distance between $C_{i,j}$ and $C_{k,g}$ on the checkerboard is computed by
(13)$$\begin{array}{c} {∥C_{i,j}-C_{k,g}∥}_{2}=l\sqrt{{\left(k-i\right)}^{2}+{\left(g-j\right)}^{2}}. \end{array}$$

The objective function for the Euclidean distance can be designed as follows:(14)$$\begin{array}{c} F_{1}=\frac{1}{{\left(NM\right)}^{2}}\sum_{i=0}^{N-1}\sum_{j=0}^{M-1}\left(\sum_{k=0}^{N-1}\sum_{g=0}^{M-1}\left|{∥P_{i,j}-P_{k,g}∥}_{2}-{∥C_{i,j}-C_{k,g}∥}_{2}\right|\right) \end{array}$$
where $P_{i,j}$ indicates the reconstructed 3D point corresponding to $C_{i,j}$ using the GU-RSRT model. This function describes the 3D reconstruction error of the Euclidean distance between th checkerboard corners. The red line in Figure 3a illustrates the Euclidean distance between $C_{i,j}$ and $C_{k,g}$.

The horizontal edges of the checkerboard are perpendicular to the vertical edges. Then, the objective function for perpendicularity can be designed as
(15)$$\begin{array}{c} F_{2}=\frac{1}{NM}\sum_{i=0}^{N-1}\left(\sum_{j=0}^{M-1}\frac{\left|\vec{P_{i,0}P_{i,M-1}}\cdot \vec{P_{0,j}P_{N-1,j}}\right|}{\left|\vec{P_{i,0}P_{i,M-1}}\right|\cdot \left|\vec{P_{0,j}P_{N-1,j}}\right|}\right). \end{array}$$

This objective function expresses that the red vector is perpendicular to the blue vector in Figure 3b.

The final objective function is a linear combination of $F_{1}$ and $F_{2}$
(16)$$\begin{array}{c} F=\omega_{1}{\bar{F}}_{1}+\omega_{2}{\bar{F}}_{2} \end{array}$$
where ${\bar{F}}_{1}$ is the normalized $F_{1}$ and ${\bar{F}}_{2}$ is the normalized $F_{2}$. Then, $\omega_{1}$ and $\omega_{2}$ directly show the importance of the Euclidean distance and perpendicularity relationships. A differential evolution algorithm is employed to solve this optimization problem (Equation (Equation 12)).

### 3.3. Marker-Based Self-Calibration

Instead of the checkerboard mentioned in the UMMR method, the structured markers embedded on the sensor surface can provide the ground truth of relative geometric features. A Marker-Based Self-Calibration (MBSC) method is proposed for specific GelStereo-type sensors. These sensors should have curved refracting surfaces, curved sensor surfaces, and markers with known structures.

Unlike the checkerboard with a planar structure, markers are distributed in 3D space. The Euclidean distance between markers is mainly used in self-calibration. In order to improve the computational efficiency without loss of geometric constraints, the voxel downsampling method is employed to downsample all markers into a few key markers. The objective function is designed as follows:(17)$$\begin{array}{c} F=\frac{1}{S^{2}}\sum_{i=1}^{S}\left(\sum_{j=1}^{S}\left|{∥P_{i}-P_{j}∥}_{2}-{∥Q_{i}-Q_{j}∥}_{2}\right|\right) \end{array}$$
where *S* is the number of markers after downsampling and $Q_{i}$ refers to the reference 3D points of $P_{i}$. Compared to the UMMR method, the marker-based self-calibration method is simple and convenient. Moreover, it can be performed every time the sensor starts up, and is able to deal with the 3D precision loss caused by the slight change of refractive indices over time.

## 4. Experiments: Design and Setup

### 4.1. Experiments Design

In order to verify the effectiveness of the proposed tactile 3D reconstruction pipeline, including the GU-RSRT model and the refractive calibration methods, we carried out the following experiments on several sensor platforms.

**Quantitative experiments on 3D reconstruction.** First of all, we quantitatively evaluate the accuracy of the proposed tactile 3D reconstruction pipeline using ground truth 3D points obtained from high-precision measuring instruments. Specifically, the 3D reconstruction errors on four different GelStereo-type sensors with various refracting surfaces and sensor surfaces were evaluated using the Mean Absolute Error (MAE) in the X, Y, Z directions and the Mean Euclidean Distance Error (MEDE) between the reconstructed 3D points and the ground truth. Moreover, on these sensor platforms we analyzed the reconstruction errors of 3D points with different contact depths and regions.

**Method comparison experiments.** In addition to the methods proposed in this paper, two other commonly used 3D reconstruction methods were used to ensure a comprehensive evaluation.

Traditional Triangulation Method (TTM): Without considering multi-medium refraction, the traditional triangulation method was applied using binocular camera parameters calibrated in the air.Camera Parameters Absorption Method (CPAM): The 3D reconstruction errors caused by multi-medium refraction can be absorbed by the camera parameters to a certain extent [40,41]. In practice, we calibrated the binocular camera using checkerboard images taken on the sensor’s transparent gel surface, then used triangulation.GU-RSRT+UMMR: The GU-RSRT model with parameters calibrated through the universal multi-medium refractive calibration method was applied to GelStereo-type sensors for tactile 3D reconstruction.GU-RSRT+MBSC: The GU-RSRT model with parameters calibrated through marker-based self-calibration method was applied to GelStereo-type sensors for tactile 3D reconstruction.

**Ablation studies.** In-depth ablation studies on the UMMR calibration method were carried out to study the importance of each relative geometric feature.

GU-RSRT+UMMR ($F_{1}$): Only the objective function for the Euclidean distance is used in UMMR calibration.GU-RSRT+UMMR ($F_{2}$): Only the objective function for perpendicularity is used in UMMR calibration.

### 4.2. Platform

#### 4.2.1. Sensor Platform

As shown in Figure 4, four GelStereo-type sensors (including GelStereo Tip, GelStereo Palm2.0, GelStereo Palm1.0, and GelStereo BioTip) with different sizes, sensor surfaces, and refracting surfaces are employed to carry out the experiments. The first row in Figure 4 diagrams the imaging system of each sensor in detail. The parameter sets which require calibration are listed in Table 2.

In these four sensors, the rays propagate in the air before entering the camera. The refractive index of air $\mu_{0}$ is 1, which is exempt from calibration. In the imaging system of GelStereo Tip, GelStereo Palm2.0, and GelStereo BioTip, the rays undergo two refractions. In the imaging system of GelStereo Palm1.0, the rays undergo three refractions, where mediums 1 and 3 are the same.

The refracting surface coordinate system and left camera coordinate system are shown in Figure 4, denoted by {S} and $\left\{C_{L}\right\}$, respectively. The equation of refracting surface $f_{i}$ in {S} is known. The transformation between {S} and $\left\{C_{L}\right\}$ is unknown, which contains structure parameters to be calibrated. In the GelStereo Tip and GelStereo Palm2.0, $d_{xy}$ indicates the distance between the left camera optical center and the XOY-plane of {S}. The direction of the z-axis of {S} expressed in $\left\{C_{L}\right\}$ is $\left(z_{x},z_{y},\sqrt{1-z_{x}^{2}-z_{y}^{2}}\right)$, known as the normal vector of the flat refracting surface. The refracting surfaces of the GelStereo Palm1.0 are hemispherical. As a result, the orientation parameters have no effect on the ray paths, and only translation is concerned; here, $\left(d_{x},d_{y},d_{z}\right)$ expresses the origin of {S} in $\left\{C_{L}\right\}$, i.e., the translation vector between {S} and $\left\{C_{L}\right\}$. Different from the GelStereo Palm1.0, the GelStereo BioTip sensor has multi-curvature refracting surfaces. Except for the translation vector, the direction vector of the x-axis of {S} in $\left\{C_{L}\right\}$ is denoted by $\left(\sqrt{1-x_{y}^{2}-x_{z}^{2}},x_{y},x_{z}\right)$.

#### 4.2.2. Ground Truth Collection Platform

A platform for collecting binocular tactile image pairs and corresponding ground truth 3D points was needed for evaluation. As shown in Figure 5a, a 3D Computer Numerical Control (CNC) linear guide was employed to generate high-precision 3D positions. A thin probe with a black dot on its tip was equipped on the tool side of 3D CNC linear guide. We used this probe tip to specify 3D positions in the workspace of linear guide. The GelStereo-type sensor without markers and coating layer was fixed on the workbench. Then, the probe was driven by the linear guide to press on the transparent gel surface of the sensor. At each sampled position, the 3D position of the probe tip was read from the linear guide, with the black dot on the probe tip being projected on the left and right image planes. As shown in Figure 5b,c, the small black dot on the probe can be clearly seen on the tactile images; to detect them, we used the blob detection algorithm in OpenCV (https://docs.opencv.org/4.x/d0/d7a/classcv_1_1SimpleBlobDetector.html, accessed on 23 February 2023). Considering the pixel position of the black dot on the images, the estimated 3D position was computed using the proposed pipeline. The 3D readings from the linear guide were converted to the left camera coordinate system using a transformation matrix, which was obtained by ArUco-based pose estimation [42] and the Iterative Closest Point (ICP) algorithm [43]. The converted 3D points are the ground truth of tactile 3D reconstruction.

### 4.3. Implementation Details

#### 4.3.1. UMMR Calibration

UMMR calibration was conducted during sensor fabrication, specifically, before painting markers and the coating layer. First, checkerboard images on the sensor surface were captured for refractive calibration using the UMMR method. Specifically, we pasted a checkerboard pattern on the flat surface of the 3D-printed calibration board. We manually pressed this calibration board onto the transparent gel surface, as shown in Figure 6a. The binocular tactile images in Figure 6b were recorded at the same time. Note that the checkerboard must be in full contact with the gel surface to ensure that all rays from the checkerboard directly enter the gel layer. The checkerboard was pressed at different positions with various poses in order to cover the sensor surface as much as possible. In addition, we chose different checkerboards for each sensor. The main principle was to select as large as possible a checkerboard pattern while ensuring complete contact with the gel surface. In practice, checkerboards with 5×4 grids with 1 mm edge length, 10×8 grids with 2 mm edge length, 10×6 grids with 1 mm edge length, and 10×6 grids with 1 mm edge length were used for the GelStereo Tip, GelStereo Palm2.0, GelStereo Palm1.0, and GelStereo BioTip, respectively. After collecting the checkerboard images, we eliminated the distortion of these images, then detected the checkerboard corners on the image plane.

Then, the optimization problem was formulated based on Equation (Equation 16) for refractive calibration. In this paper, min-max normalization was employed for the objective function value of $F_{1}$ and $F_{2}$ on the pre-set parameter ranges. The weights $\left(\omega_{1},\omega_{2}\right)$ in Equation (Equation 16) were set as (0.6, 0.4) for the GelStereo Tip, (0.8, 0.2) for the GelStereo Palm2.0, (0.5, 0.5) for the GelStereo Palm1.0, and (0.2, 0.8) for the GelStereo BioTip.

Finally, this optimization problem was solved using a differential evolution algorithm. The bounds of parameters are listed in Table 3, and were set according to prior knowledge, such as the material properties and sensor structure.

#### 4.3.2. Data Collection for Evaluation

Using the platform in Figure 5, points on the sensor surface were sampled to evaluate the 3D reconstruction errors. These sampling points are illustrated on the image plane with red dots, as shown in Figure 7. The probe was programmed to press at these sampling points with multiple depths. In practice, on the GelStereo Tip sensor, each sampling surface point was pressed seven times, with the depth from 0 mm to 3 mm in 0.5 mm intervals. On the GelStereo Palm2.0 sensor, each sampling surface point was pressed eight times, with the depth from 0 mm to 3.5 mm in 0.5 mm intervals. On the GelStereo Palm1.0 sensor, each sampling surface point was pressed eight times, with the depth from 0 mm to 2.8 mm in 0.4 mm intervals. On the GelStereo BioTip sensor, each sampling surface point was pressed four times, with the depth from 0 mm to 1.2 mm in 0.4 mm intervals.

To evaluate the 3D reconstruction errors of different regions on the sensor surface, these sampling points were divided into several groups, which are depicted by yellow lines and numbers in Figure 7; the larger the number, the outer the points.

## 5. Experimental Results

**3D reconstruction accuracy.** The refractive calibration results and 3D reconstruction errors of GelStereo Tip sensor, GelStereo Palm2.0 sensor, GelStereo Palm1.0 sensor, and GelStereo BioTip sensor are shown in Table 4, Table 5, Table 6 and Table 7, respectively. The proposed 3D reconstruction pipeline achieves mean Euclidean distance errors of 0.183 mm on the GelStereo Tip, 0.256 mm on the GelStereo Palm2.0, 0.264 mm on the GelStereo Palm1.0, and 0.328 mm on the GelStereo BioTip, indicating the feasibility of the proposed method on GelStereo-type sensors with different structures.

**Methods comparison.** In Figure 8, we visualize the tactile 3D point clouds obtained by different 3D reconstruction methods. The 3D reconstruction errors are shown in Table 4, Table 5, Table 6 and Table 7. We find that 3D reconstruction using the proposed GU-RSRT model and refractive calibration method outperforms the other methods. On the GelStereo Tip, GelStereo Palm1.0, and GelStereo BioTip sensors, the MEDE of TTM is about 10 times that of the proposed method, while on the GelStereo Palm2.0 the performance of TTM is even worse. As shown in Figure 8, the 3D point clouds reconstructed using TTM are severely distorted on the GelStereo Palm1.0 and GelStereo BioTip sensors with curved refracting surfaces. In addition, the CPAM, commonly used in underwater environments, performs poorly on GelStereo-type sensors, and the MEDE is about 10 mm. One possible reason is that the refracting surface in GelStereo-type sensor imaging systems is farther away from the camera optical center compared to underwater scenarios, leading to difficulties in absorbing the refraction effects with these camera parameters.

**Self-calibration.** The results of marker-based self-calibration and the 3D reconstruction errors of the GelStereo Palm1.0 and GelStereo BioTip sensors are shown in Table 6 and Table 7, respectively. We find that the MBSC method performs better than the UMMR calibration method on the GelStereo Palm1.0 and GelStereo BioTip. Specifically, the MEDE is reduced by about 0.03 mm. This is probably because the relative geometric features in 3D space (i.e., markers on curved surface) are more helpful for calibration than those in 2D plane (i.e., checkerboards). Compared to the UMMR calibration method, the MBSC method is more accurate and convenient. Therefore, marker-based self-calibration is the preferred calibration method for these specific GelStereo-type sensors, which have curved refracting surfaces, curved sensor surfaces, and markers with known structures.

In addition, we applied this marker-based self-calibration method to the GelStereo Tip and GelStereo Palm2.0 sensor; however, the performance was not satisfactory, as shown in Table 4 and Table 5. According to [44], the 3D reconstruction errors of the GelStero Tip and GelStereo Palm2.0 caused by multi-medium refraction are mainly distributed in the z-axis direction, which is due to the refracting surfaces in the GelStero Tip and GelStereo Palm2.0 being flat and the normal of the refracting surface almost parallel to the z-axis. Therefore, the relative geometric information in the z-axis is significant for calibration. On the GelStereo Tip, the sensor surface is flat, and the 3D coordinates of markers are almost consistent on the z-axis. Because of this, the self-calibration method based on marker distance barely works on the GelStereo Tip sensor. On the GelStereo Palm2.0, the performance of the marker-based self-calibration method might be improved by a specific marker sampling method that pays more attention to the z-axis.

**3D reconstruction errors with different contact depths and regions.** We further studied the reconstruction accuracy of the proposed method on different sensor platforms for different contact depths and contact regions in order to evaluate its robustness. The error distributions at different contact depths are illustrated using a violin plot in Figure 9, which indicates that the 3D reconstruction accuracy of GelStereo-type sensors using the proposed method is almost independent of contact depth. In addition, the 3D errors at different contact regions are depicted in Figure 10. The region numbers correspond to the numbers in Figure 7. It can be seen that the reconstruction errors of the outer points are larger than those of the middle points on the sensor surface. Specifically, the MEDE of the outermost points is about 1.4 times of the mean value. One possible reason for this is that the projection of the black dot (on the probe) on the image plane is stretched into an ellipse-like shape, when the probe presses at the outer regions. Inevitably, the commonly used blob detection algorithm introduces errors to the center of black dot, which affects the accuracy of 3D reconstruction. Moreover, on the GelStereo BioTip sensor we find that the errors in regions 4 and 5 are significantly larger. In Figure 8d, several green points in the positive X direction protrude from the sensor surface. This might be caused by the accuracy loss of the refracting surface equation. To solve this problem, it might be possible to either improve the production accuracy of the transparent supporting shell where refraction occurs or correct the equation of refracting surface through calibration.

**Ablation studies.** The results of ablation studies on the UMMR calibration method are demonstrated in Table 8. The experimental results show that the objective function combining Euclidean distance and perpendicularity works better than a single objective function with Euclidean distance $F_{1}$ or perpendicularity $F_{2}$ on all these GelStereo-type sensors. Moreover, we find that the UMMR calibration method with objective function $F_{1}$ achieves a MEDE of less than 0.6 mm on these sensors, which is not much worse than the full UMMR method. This finding shows the importance and indispensability of Euclidean distance features, especially on the GelStereo Tip, GelStereo Palm2.0, and GelStereo Palm1.0 sensors. For achieving better constraints on spatial geometry, the perpendicular features are complementary to Euclidean distance features.

## 6. Discussion

In GelStereo-type sensor imaging systems, the shapes of refracting surfaces play an important role in ray tracing. Although the refracting surface function depends on sensor design, the manufacturing process (such as 3D printing, laser cutting, etc.) of the transparent supporting plate might bring errors into this function, especially on the curved supporting plate in the GelStereo Palm1.0 and GelStereo BioTip sensors. To solve this problem, a method for correcting the function of the refracting surface should be integrated into the refractive calibration process. In this way, the precision of tactile 3D reconstruction can be further improved in GelStereo-type sensors.

As mentioned in Section 5, with a flat refracting surface the relative geometric information in the surface normal direction needs special attention during calibration. For marker-based self-calibration with high precision and computational efficiency, an algorithm generating marker pairs to compute the Euclidean distance is required, which solves the problem of effectively representing the geometric features with fewer markers on various GelStereo-type sensor platforms.

## 7. Conclusions

In this paper, we present a universal Refractive Stereo Ray Tracing model for GelStereo-type sensors to model the tactile 3D reconstruction under multi-medium light refraction. In addition, a Universal Multi-Medium Refractive (UMMR) calibration method is proposed to obtain the refractive and structural parameters in the GU-RSRT model, in which relative geometric features on checkerboards are employed to build an optimization problem for calibration. In addition, a self-calibration method based on structured markers on the sensor surface is provided for specific GelStereo-type sensors.

Extensive calibration and evaluation experiments are conducted on four different GelStereo-type sensors with various structure designs. The experimental results show that the proposed refractive calibration method can obtain reasonable parameters of the GU-RSRT model and the 3D reconstruction error of the mean Euclidean distance error is less than 0.35 mm, which outperforms the other 3D reconstruction methods. In addition, the accuracy of the marker-based self-calibration method is slightly better than the UMMR calibration method on GelStereo-type sensors with curved refracting surfaces. The self-calibration method has great potential in improving calibration efficiency and sensor service life. Moreover, our experimental results show the robustness of the proposed 3D reconstruction pipeline with different contact depths and regions.

The feasibility of the proposed tactile 3D reconstruction pipeline is fully demonstrated in this paper. Its practical application scenario is visuotactile sensing (especially high-precision 3D contact geometry measurement) based on binocular cameras and undergoing multi-medium light refraction. With high-precision sensing capability, GelStereo-type sensors and other similar visuotactile sensors could provide more possibilities for robots to achieve rich-contact and dexterous manipulation. In the future, we intend to further improve the 3D reconstruction performance of GelStereo-type sensors and apply GelStereo-type sensors to robotic perception and manipulation tasks.

## Figures and Tables

**Figure 1 sensors-23-02675-f001:**
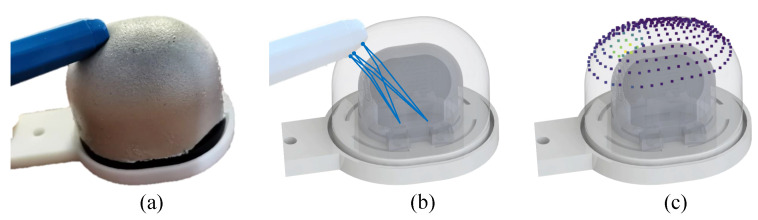
Diagram of tactile 3D reconstruction based on ray tracing: (**a**) real GelStereo-type sensor contacted with a pen; (**b**) ray-tracing diagram (the blue lines indicate light ray paths inside the sensor undergoing multi-medium refraction); (**c**) reconstructed 3D point cloud (the points in light color denote the contact area).

**Figure 2 sensors-23-02675-f002:**
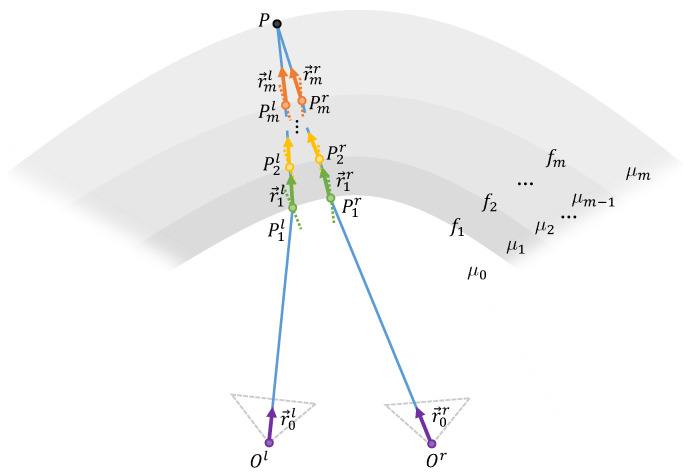
Diagram of the GU-RSRT model.

**Figure 3 sensors-23-02675-f003:**
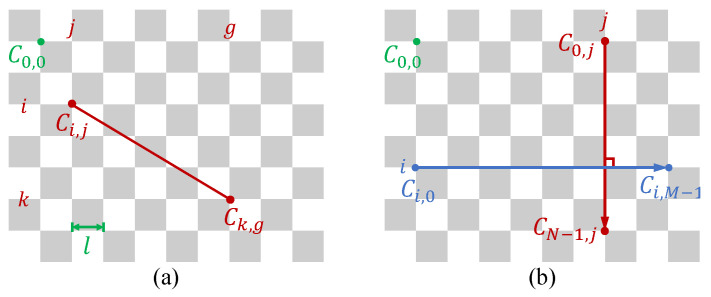
The relative geometric features on checkerboard: (**a**) the Euclidean distance feature and (**b**) the perpendicularity feature.

**Figure 4 sensors-23-02675-f004:**
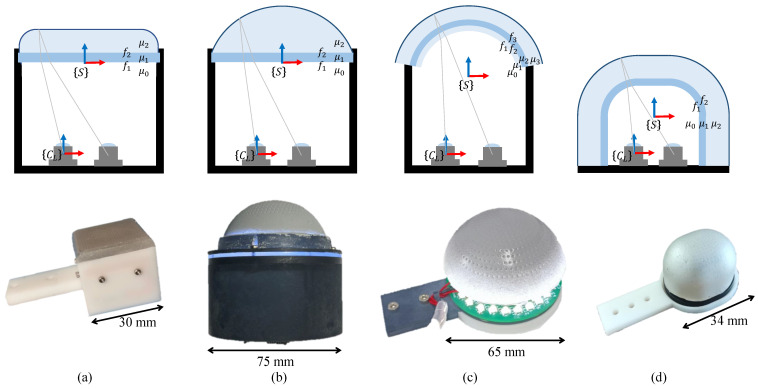
The GelStereo-type sensors. The first row shows a diagram of sensor imaging systems with multi-medium refraction. The second row shows pictures of real-world sensors. (**a**) The GelStereo Tip sensor. (**b**) The GelStereo Palm sensor with flat refracting surface (GelStereo Palm2.0). (**c**) The GelStereo Palm sensor with hemispherical refracting surface (GelStereo Palm1.0) [37]. © [2023] IEEE. Reprinted, with permission, from [37]. (**d**) The GelStereo BioTip sensor.

**Figure 5 sensors-23-02675-f005:**
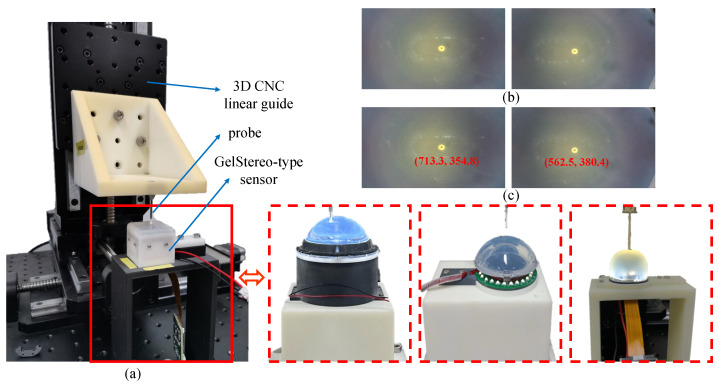
The platform for ground truth 3D points collection. (**a**) The 3D CNC linear guide with GelStereo-type sensors. (**b**) The left and right tactile image pairs when the probe presses on the sensor surface. The black dot on the probe tip can be clearly seen in the images. (**c**) The blob detection results of (**b**) are shown with a red dot on the image. The red tuples indicate the pixel positions.

**Figure 6 sensors-23-02675-f006:**
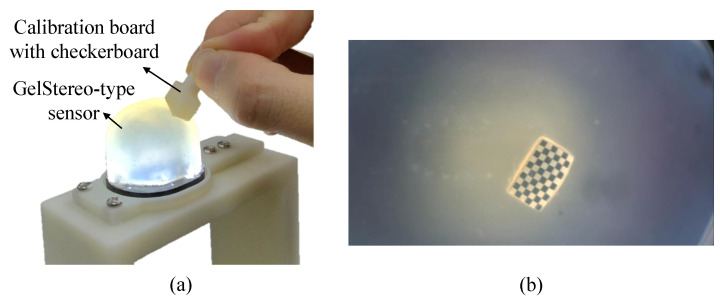
Checkerboard images collection on GelStereo-type sensor. (**a**) The image collection scene. (**b**) Right tactile image of the checkerboard.

**Figure 7 sensors-23-02675-f007:**
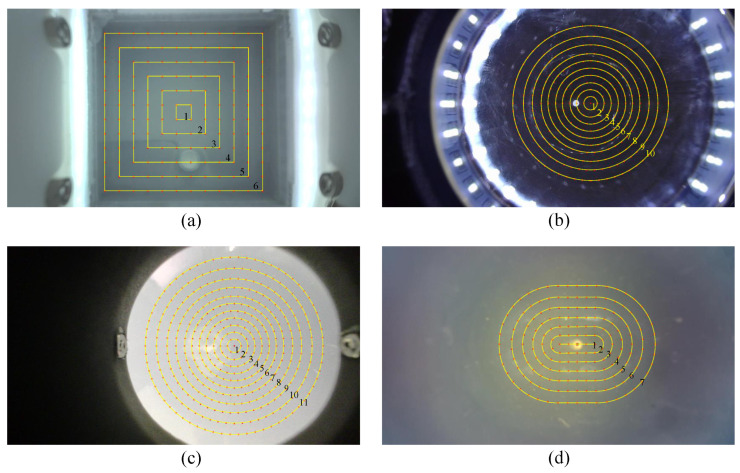
Illustration of sampling points on sensor surfaces: (**a**) GelStereo Tip, (**b**) GelStereo Palm2.0, (**c**) GelStereo Palm1.0, (**d**) GelStereo BioTip sensor.

**Figure 8 sensors-23-02675-f008:**
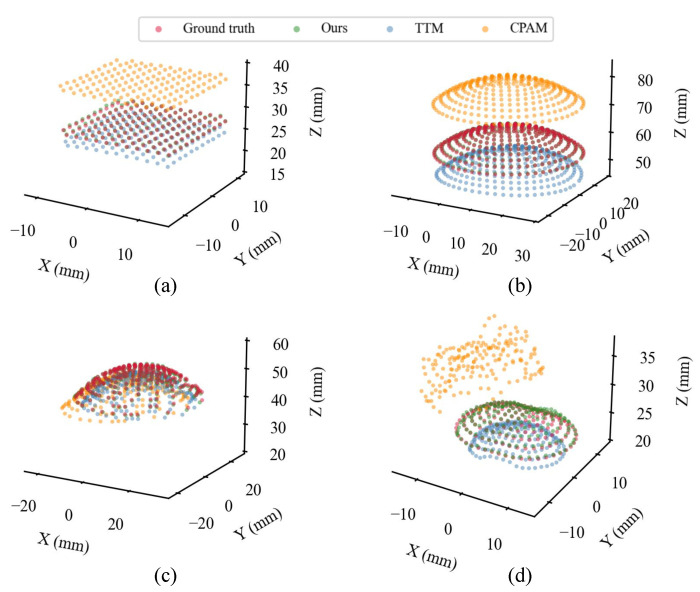
Visualization of tactile 3D point clouds on various sensors using different methods: (**a**) GelStereo Tip, (**b**) GelStereo Palm2.0, (**c**) GelStereo Palm1.0, (**d**) GelStereo BioTip.

**Figure 9 sensors-23-02675-f009:**
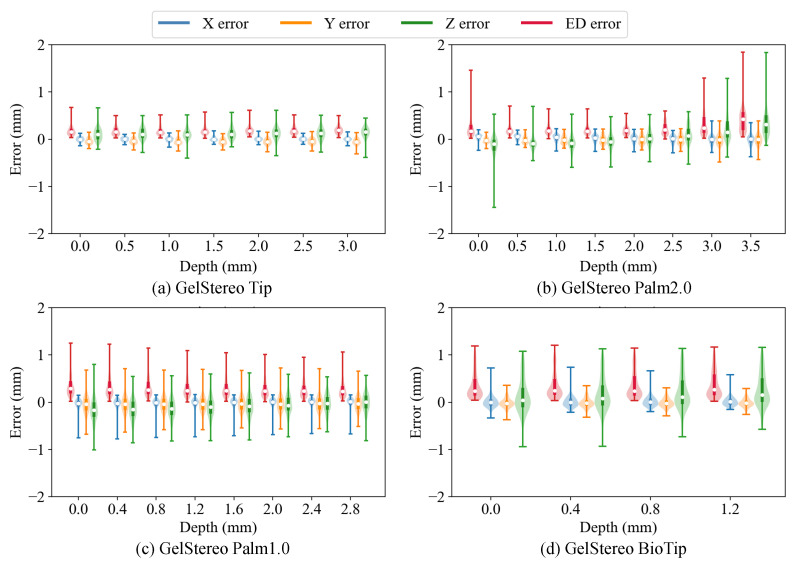
Violin plot showing the distribution of 3D reconstruction errors at different contact depths. The X error, Y error, Z error, and ED error denote the error on the X-axis, Y-axis, Z-axis, and Euclidean distance between the reconstructed 3D points and the ground truth. The white dots indicate medians.

**Figure 10 sensors-23-02675-f010:**
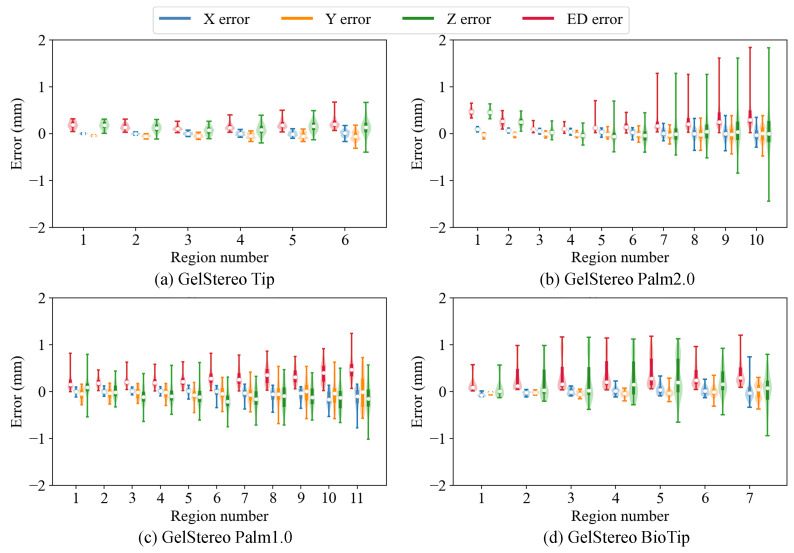
Violin plot showing the distribution of 3D reconstruction errors at different contact regions. The X error, Y error, Z error, and ED error denote the error on the X-axis, Y-axis, Z-axis, and Euclidean distance between the reconstructed 3D points and the ground truth. The white dots indicate medians.

**Table 1 sensors-23-02675-t001:** Notation for the GU-RSRT model.

Symbols	Descriptions
l,r	the ray from left or right camera optical center (superscripts)
*i*	the index of media, i∈{0,1,2,⋯,m} (subscripts)
L,R	the left or right camera coordinate system (subscripts/superscripts)
$O^{*}\in R^{3}$	the camera optical center
${\vec{r}}_{*}\in R^{3}$	the direction vector of the ray (unit vector)
${\vec{n}}_{*}\in R^{3}$	the normal vector of refracting surface (unit vector)
$P_{*}\in R^{3}$	the intersections of the ray and refracting surfaces (3D points)
$\mu_{*}\in R^{1}$	refractive indices

**Table 2 sensors-23-02675-t002:** The parameter sets to be calibrated in each sensor.

Sensors	Parameters to Calibrate
GelStereo Tip	$\left\{\mu_{1},\mu_{2},d_{xy},z_{x},z_{y}\right\}$
GelStereo Palm2.0	$\left\{\mu_{1},\mu_{2},d_{xy},z_{x},z_{y}\right\}$
GelStereo Palm1.0	$\left\{\mu_{1}=\mu_{3},\mu_{2},d_{x},d_{y},d_{z}\right\}$
GelStereo BioTip	$\left\{\mu_{1},\mu_{2},d_{x},d_{y},d_{z},x_{y},x_{z}\right\}$

**Table 3 sensors-23-02675-t003:** Parameter ranges of the GelStereo-type sensors used for optimization.

Parameters	GelStereo Tip	GelStereo Palm2.0	Parameters	GelStereo Palm1.0	GelStereo BioTip
$\mu_{1}$	[1.21,1.61]	[1.21,1.61]	$\mu_{1}$ ($\mu_{3}$)	[1.21,1.61]	[1.29,1.69]
$\mu_{2}$	[1.21,1.61]	[1.21,1.61]	$\mu_{2}$	[1.29,1.69]	[1.21,1.61]
$d_{xy}$ (mm)	[19.0,25.0]	[30.0,36.0]	$d_{x}$ (mm)	[6.3,10.3]	[2.2,4.2]
$z_{x}$	[−0.2,0.2]	[−0.2,0.2]	$d_{y}$ (mm)	[−1.0,1.0]	[−1.0,1.0]
$z_{y}$	[−0.2,0.2]	[−0.2,0.2]	$d_{z}$ (mm)	[20,26]	[12.5,16.5]
			$x_{y}$	/	[−0.1,0.1]
			$x_{z}$	/	[−0.1,0.1]

**Table 4 sensors-23-02675-t004:** The calibration results and 3D reconstruction errors of the GelStereo Tip sensor. The unit of $d_{xy}$, X MAE, Y MAE, Z MAE, and MEDE is mm.

Methods	$\mu_{1}$	$\mu_{2}$	$d_{\mathrm{xy}}$	$z_{x}$	$z_{y}$	X MAE	Y MAE	Z MAE	MEDE
TTM	/	/	/	/	/	0.070	0.098	1.635	1.642
CPAM	/	/	/	/	/	0.157	0.167	9.523	9.527
GU-RSRT+UMMR (ours)	1.468	1.405	21.016	−0.0111	0.0157 0.038	0.068	0.146	0.183	
GU-RSRT+MBSC (ours)	1.604	1.599	21.510	0.0611	0.0723	0.134	0.083	0.552	0.584

**Table 5 sensors-23-02675-t005:** The calibration results and 3D reconstruction errors of GelStereo Palm2.0 sensor. The unit of $d_{xy}$, X MAE, Y MAE, Z MAE, and MEDE is mm.

Methods	$\mu_{1}$	$\mu_{2}$	$d_{\mathrm{xy}}$	$z_{x}$	$z_{y}$	X MAE	Y MAE	Z MAE	MEDE
TTM	/	/	/	/	/	0.146	0.141	7.624	7.629
CPAM	/	/	/	/	/	0.343	1.011	17.795	17.828
GU-RSRT+UMMR (ours)	1.443	1.369	33.935	−0.0003	−0.0058	0.076	0.088	0.200	0.256
GU-RSRT+MBSC (ours)	1.680	1.593	30.002	0.0110	0.0098	0.170	0.172	7.303	7.307

**Table 6 sensors-23-02675-t006:** The calibration results and 3D reconstruction errors of GelStereo Palm1.0 sensor. The unit of $d_{x}$, $d_{y}$, $d_{z}$, X MAE, Y MAE, Z MAE, and MEDE is mm.

Methods	$\mu_{1}=\mu_{3}$	$\mu_{2}$	$d_{x}$	$d_{y}$	$d_{z}$	X MAE	Y MAE	Z MAE	MEDE
TTM	/	/	/	/	/	0.692	0.772	2.101	2.470
CPAM	/	/	/	/	/	6.120	0.161	3.695	7.172
GU-RSRT+UMMR (ours)	1.324	1.470	8.259	−0.126	23.496	0.086	0.154	0.193	0.291
GU-RSRT+MBSC (ours)	1.379	1.522	7.503	0.149	23.793	0.105	0.123	0.174	0.264

**Table 7 sensors-23-02675-t007:** The calibration results and 3D reconstruction errors of GelStereo BioTip sensor. The unit of $d_{x}$, $d_{y}$, $d_{z}$, X MAE, Y MAE, Z MAE, and MEDE is mm.

Methods	$\mu_{1}$	$\mu_{2}$	$d_{x}$	$d_{y}$	$d_{z}$	$x_{y}$	$x_{z}$	X MAE	Y MAE	Z MAE	MEDE
TTM	/	/	/	/	/	/	/	0.810	0.994	2.816	3.224
CPAM	/	/	/	/	/	/	/	9.263	1.799	8.477	12.871
GU-RSRT+UMMR (ours)	1.509	1.373	3.450	−0.115	14.218	−0.0034	0.0304	0.093	0.079	0.311	0.365
GU-RSRT+MBSC (ours)	1.594	1.450	3.576	0.003	14.823	0.0027	0.0156	0.069	0.075	0.294	0.328

**Table 8 sensors-23-02675-t008:** The results of ablation studies on the UMMR calibration method. The unit of X MAE, Y MAE, Z MAE, and MEDE is mm.

	GelStereo Tip				GelStereo Palm2.0			
	X MAE	Y MAE	Z MAE	MEDE	X MAE	Y MAE	Z MAE	MEDE
GU-RSRT+UMMR ($F_{1}$)	0.038	0.066	0.163	0.197	0.078	0.092	0.330	0.375
GU-RSRT+UMMR ($F_{2}$)	0.135	0.059	0.173	0.249	0.075	0.138	1.392	1.406
GU-RSRT+UMMR	0.038	0.068	0.146	0.183	0.076	0.088	0.200	0.256
	**GelStereo Palm1.0**				**GelStereo BioTip**			
	**X MAE**	**Y MAE**	**Z MAE**	**MEDE**	**X MAE**	**Y MAE**	**Z MAE**	**MEDE**
GU-RSRT+UMMR ($F_{1}$)	0.114	0.159	0.225	0.331	0.098	0.116	0.539	0.574
GU-RSRT+UMMR ($F_{2}$)	0.290	0.363	1.013	1.169	0.127	0.126	0.297	0.391
GU-RSRT+UMMR	0.086	0.154	0.193	0.291	0.093	0.079	0.311	0.365

## Data Availability

Not applicable.

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
