# Peer review of "High-Precision 3D Reconstruction Study with Emphasis on Refractive Calibration of GelStereo-Type Sensors"

_sensors, 2023, doi:10.3390/s23052675_

Round 1

Reviewer 1 Report

Review results

In this paper, a general GelStereo-type sensor refraction stereoscopic ray tracing model is proposed for haptic 3D reconstruction under multi-medium light refraction. The scaling optimization problem is established by using the relative geometric features of the checkerboard. In addition, a self-calibration method based on the surface structure markers of the GelStereo-type sensor is provided for specific Gelstereo-type sensors. Through analyzing the experimental results, the feasibility of the proposed tactile 3D reconstruction pipeline is fully demonstrated. The full paper has a regular interface, clear writing ideas, full analysis and discussion of experimental results, even clear work plan for the future.

There are also the following questions:

1.The description of the practical application scenario of this research is not obvious. What is the application value or scenario of this technology? Theoretical or practical application value?

2. What does the black probe in Figure 4 do? What do the two pictures in Figure 4 (b) and (c) represent respectively? What's the difference of the two pictures?

3. The first three parts of the article account for a little more of the whole article

The fourth part of the expression of platform is a little length.

Reviewer 2 Report

Author provide an idea of optical calibration and comparison. 

Modify/Replace the figure1 to better one. In the beginning, the figure1 makes us hard to grasp the concept and background of idea. An appropriate figure such as Fig3 or photo will provide better start-up to readers.

Has author measured those optical information of sensors? From what I understand, radius of surfaces, thickness of mediums, refractive indices, and position of cameras are significant for verifying the optical methods and comparisons.

Check all sentences. For example: left and right ray -> rays

Remove overused bullets to avoid report style. 

Change the bulleted question sentences to statements, and put them at the head of the paragraphs. 

Use 2.1, 2.2, ... instead of bolds.

Avoid repeated statements about accomplishments.
